# Design of Nanohydroxyapatite/Pectin Composite from *Opuntia Ficus-Indica* Cladodes for the Management of Microbial Infections

**DOI:** 10.3390/polym14204446

**Published:** 2022-10-20

**Authors:** N. Saidi, K. Azzaoui, M. Ramdani, E. Mejdoubi, N. Jaradat, S. Jodeh, B. Hammouti, R. Sabbahi, A. Lamhamdi

**Affiliations:** 1Laboratory of Applied Chemistry and Environment (LCAE), Department of Chemistry, Faculty of Sciences, University Mohamed I, Oujda P.O. Box 717, Morocco; 2Department of Pharmacy, Faculty of Medicine and Health Sciences, An-Najah National University, Nablus P.O. Box 7, Palestine; 3Department of Chemistry, An-Najah National University, Nablus P.O. Box 7, Palestine; 4Laboratory of Plant Biotechnology, Higher School of Technology, University of Ibn Zohr Quartier 25 Mars, Laayoune P.O. Box 3007, Morocco; 5Laboratory of Applied Sciences, National School of Applied Sciences (ENSAH), Abdelmalek Essaadi University, Tetouan P.O. Box 2222, Morocco

**Keywords:** antimicrobial activity, cladode, hydroxyapatite, *Opuntia ficus-indica*, polymer composites, nanoparticles, pectin, prickly pear

## Abstract

Hydroxyapatite (HAp) attracts interest as a biomaterial for use in bone substitution or allografts. In the current work, biomaterial nanocomposites based on HAp and pectin were synthesized by using the double decomposition method, which involved using pectin extracted from fresh cladodes of the prickly pear, *Opuntia ficus-indica*. The crystallinity, purity, and several analytical techniques like Fourier transform infrared spectroscopy, X-ray diffraction, and scanning electron microscopy were used to understand the surface’s shape. The results revealed that the produced HAp/pectin nanoparticles are pure, spherical, and amorphous. The spectroscopic data indicated a substantial interaction between HAp and pectin, specifically between Ca (II) and pectin hydroxyl and carboxyl groups. The presence of pectin showed a noticeable influence on the prepared nanocomposite texture and porosity. We further assess the antibacterial and antifungal activity of the developed nanocomposite against a number of pathogenic bacteria and fungi, evaluated by the well diffusion method. In the absence of pectin, the XRD analysis revealed that the HAp nanoparticles had 10.93% crystallinity. When the pectin concentration reached 10 wt.%, it was reduced to approximately 7.29%. All synthesized nanocomposites demonstrated strong antimicrobial activity against both Gram-positive (*Staphylococcus aureus* and *Bacillus cereus*) and Gram-negative (*Escherichia coli* and *Pseudomonas aeruginosa*) bacteria in addition to various fungi (e.g., *Aspergillus fumigatus*, *Penicillium funiculosum*, and *Trichoderma viride*). This study endorses the HAp/Pectin nanocomposite as an efficient antimicrobial material for biomedical advanced applications.

## 1. Introduction

The industrial applications of medicinal plants, especially from national flora, are of great benefit to the pharmaceutical industry and have a definite economic impact on several countries. The prickly pear, *Opuntia ficus-indica* (L.), has many valuable uses for the manufacturing of agrifood, cosmetic, and pharmaceutical products [1,2]. This cactus plant cell wall is made up of several layers of macromolecules; among them are cellulose, hemicellulose, pectin, and lignin [3]. Pectin is an organic substance that is found in the middle lamella of the primary cell wall of many plants. It gives the plant’s parts rigidity and strength. The water-soluble polysaccharide extracted from *O. ficus-indica* cladodes contains mainly pectin [4]. The basic structure of pectin is a linear central chain composed of α-D-galacturonic acid units linked by α-(1→4) glycosidic bonds. The unit is partially acetylated and methylated at O-2 or O-3 [5]. Pectin could be extracted from various types of fruits’ seeds such as apple pomace, citrus peel, beet pulp [6], banana peel, and murta fruit [7].

Pectin is an important commercial ingredient that is widely used for the manufacturing of antidiarrheal medications. It is also used in the food industry as a thickener, gelling agent, emulsifier, and stabilizer. It is mainly used in fruit juices, jellies, and jams, in cooking fruit preparations, fruit drink concentrates, desserts, dairy products, and delicatessen products [8,9]. 

Pectin is known as the miracle polymer due to its excellent biodegradability and biocompatibility [10] with an unlimited number of applications, such as in the preparation of pharmaceutical and cosmetic products [11], as well as in enzymatic immobilization [12]. Moreover, pectin can be consumed as a food component or as a dietary supplement [13]. In addition, a recent publication showed that pectin extracted from banana peels can be used as a mediator in the preparation of Hydroxyapatite (HAp) [Ca_10_(PO_4_)_6_(OH)_2_] nanoparticles [14]. Pectin can be also used to synthetize copolymers based on chitosan and/or HAp [8].

HAp is the principal component of mankind’s hard tissues, such as living teeth and bone, and it has found applications in biomedical fields. For example, nanoparticles enable homogenous resorption by osteoclasts and are employed in bone healing and replacement of injured or traumatized bone tissues [15]. Several studies showed that HAp and its composites could be valuable materials in bone tissue repair, prevent the proliferation of cancer cells, and be an efficient agent in the field of drug delivery [16]. In another study, nano-zirconium dioxide and hydroxyapatite-reinforced polyetheretherketone (HAp/ZrO_2_/PEEK) biocomposites could give bone calcification. The PEEK biocomposite with 2 wt% HAp and 5 wt% ZrO_2_ nanoparticles [17,18] displayed the highest tribological properties and bioactivity, which could help in developing potential medical materials for clinical applications [19]. Cisplatin-loaded graphene oxide/chitosan/HAp composite was used in the controlled release of cisplatin. This composite was also found to show several advantages like enhancing proliferative, adhesion effects on osteoblast-like cells’ alkaline phosphatase activities. These effects make it very special to be used for bone tissue replacement when there are bone-cancer-affected tissues [15].

The synthesis of nanoparticles of HAp with controlled sizes and morphologies is well documented in the literature. Among the reported methods of HAp preparation are hydrothermal, sol-gel, co-precipitation, ultrasonic irradiation [20], freezing [21], template assisted synthesis, and reverse microemulsion [21]. In recent years, the attention of researchers has been directed toward a natural product mediated approach for forming nanoparticles of HAp since natural products have several advantages, such as being abundant, biodegradable, cost-effective, safe, renewable, and environmentally benign [22]. In addition, the composites made from such materials are biodegradable. Among the natural materials used in this area are carbohydrates, proteins, fibers, and polysaccharides.

Polymers have been extensively used as nanocarriers in recent decades [23,24,25]. The antibacterial polymers usually obtained either by synthesis of monomeric biocide moiety and subsequent polymerization or copolymerization with another monomer or modification can also be brought by grafting of N-alkylated poly (4-vinylpyridine) quaternized polyethyleneimine and quaternary derivatives of acrylic acid onto numerous materials such as cellulose [26]. Other studies demonstrate that ZnO/Ag nanocomposites, membranes loaded with HAp-Ag_3_PO_4_ nanoparticles, and silver nanoparticle–activated carbon composite nanofiber membranes exhibit an enhanced antimicrobial activity [17,18,27].

Both HAp and pectin have the functionality required for strong interaction and compatibility represented by H-bonding. The main objective of the present study is to prepare a novel biocomposite of HAp and pectin, obtained from *O. ficus-indica*. A process for making nanoparticles is offered in this study. FT-IR, XRD, and SEM studies were used to characterize the synthetized nanocomposite. The HAP/pectin nanocomposite performed a dual function in water purification. It exhibited a high affinity for a variety of antimicrobial activity against both Gram-negative and Gram-positive bacteria, and fungi, providing a new outlook for the development of antimicrobial agents. 

## 2. Materials and Methods

### 2.1. Materials

The utilized pectin in this work was isolated from fresh *O. ficus-indica* cladodes that were collected in March 2021 from Oujda region, Morocco. HAp was synthetized in our laboratory. Calcium nitrate Ca(NO_3_)_2_.4H_2_O (99.0%), ammonium hydrogen phosphate (NH_4_)_2_HPO_4_ (99.0%), and dimethyl formamide were all purchased from Sigma-Aldrich, France, with high purity (greater than 99%), and were used as received. Purity deionized water was used in all runs. 

### 2.2. Pectin Extraction

A sample of cladode (300 g) was cut into small pieces and suspended in distilled water (100 mL) and placed in a conventional microwave (MS-23F301EFS model, Samsung, Experience Store, Rabat, Morocco) set at 2450 MHz and 700 W for 15 min. After cooling, the produced suspension was centrifuged for 20 min at 4500 rpm. The pectin-containing supernatant was filtered through a fine piece of cloth. The filtrate was diluted with absolute ethanol in a 2:3 by volume (supernatant to solvent) ratio to precipitate pectin. The precipitate was collected by filtration, washed with an ethanol–water solution (70 wt.%) to remove any leftover impurities, and dried at room temperature overnight. The solid was pulverized and stored in a glass container at room temperature. The equipment and materials used in the extraction process are shown in Figure 1.

### 2.3. Synthesis of Hydroxyapatite

A modified version of the wet chemical method [28] was used to synthesize HAp. A solution of calcium nitrate (11.76 g) in 100 mL of water was prepared at room temperature. An aqueous solution of diammonium phosphate (4.06 g) in 100 mL of water was prepared and dropped into the calcium nitrate solution over a 30 min period. Based on a stoichiometric HAp, the amount of reagents in the solution was adjusted to obtain a Ca-to-P molar ratio of 1.67. The pH of the slurry was tested during the precipitation reaction and found to be 10.5.

### 2.4. Synthesis of Hydroxyapatite/Pectin Nanoparticles 

At room temperature, an aqueous solution of pectin polymer was made and labeled as solution A. A prepared HAp sample was disseminated in dimethylformamide and labeled solution B. The two solutions were then combined. After approximately 1.5 h, an opaque milky white suspension was generated. Over a 30 min period, the temperature of the generated suspension was gradually elevated at a rate of 2 °C/min to 50 °C. To obtain a homogeneous suspension, the temperature was kept at 50 °C for 2 h. The nanoparticles in the suspension were centrifuged and washed twice with ethanol to eliminate unreacted material and potential contaminants, and to speed dry it. HAp/pectin nanocomposites were synthetized with molar ratios of 100/0, 50/50, and 90/10.

### 2.5. Analyzing Methods

Prepared HAp/pectin nanocomposites along with the used starting materials were analyzed using various spectroscopic and analytical techniques, which included the Fourier transform infrared (FT-IR) spectroscopy, X-ray diffraction analysis (XRD), and total carbon content. The FT-IR analysis was performed on a Schimadzu FT-IR 300 series analyzer (Shimadzu Scientific Instruments, Kyoto, Japan) in the 600–4000 cm^−1^ range with a resolution of 2 cm^−1^ and 128 scans. The tested composites’ KBr pellets were made by combining 0.025 g of the composite with 0.975 g of KBr. The morphology of the composite was examined using an Emission-Scanning Electron Microscopy (SEM/FEG) an SU 8020, 3.0 KV SE(U) (Hitachi, Japan).

The X-ray diffraction (XRD) data were collected using an X-ray powder diffractometer (LabxXRD-6100 Shimadzu, Shimadzu, Japan) with a Cu K radiation (40 kV, 30 mA) of 0.154 nm at an acquisition rate of 0.05 to 25°/min. Using Debye–Scherrer’s equation (Equation (1)), the peak broadening of XRD reflection data was utilized to estimate the crystallite size at a perpendicular site to the crystallographic plane [23].
(1)XS=0.9 λβcosθ
where *Xs* is the crystalline size in nm; *λ* is the wavelength acquired from the X-ray beam, which was set at roughly 0.15406 nm for Cu K radiation; *θ* is the diffraction angle (°); and *β* represents the full width at half maximum (FWHM) for various peaks under consideration (rad).

The crystallinity and size were calculated using many peaks at 26°. Equation (2) was used to calculate the crystallinity percent *Xc* of the produced HAp nanoparticles [23]:(2)Xc=0.24β 3
where *β* is the FWHM.

The TOC/TN analyzer multi-N/C 2100S duo was used to measure total organic carbon (TOC), and total carbon (TC) was discovered to reflect the carbon content in an aqueous phase effluent. *Muller–Hinton Broth* (Biokar), *Muller–Hinton Agar* (Biokar), and *Potato Dextrose Agar* (PDA) were used to test the TC.

### 2.6. Study of Antibacterial Activity

The eight bacterial strains employed in this study (Table 1) were chosen for their high pathogenicity and antibiotic resistance. They are responsible for serious infections in humans. The strains are composed of Gram-negative and Gram-positive types. The antibacterial efficacy test was done using a micro-dilution method [29]. In this study, several bacterial suspensions were brought in sterile saline with a concentration of 1.0 × 105 CFU/mL. All composites were dissolved in DMSO (5%) solution with surfactant containing 0.1% Tween 80 (*v/v*) (1 mg/mL). The tryptic soy broth (TSB) medium (100 μL) was added with an inoculum bacterial of 1.0 × 104 CFU per well. The minimum concentrations that caused no visible growth that could be detected by a binocular microscope were defined as the concentrations that completely inhibited bacterial growth. The MICs were obtained using a colorimetric microbial viability test based on the reduction in an INT dye (iodonitrotetrazolium) with a violet color and were compared to a positive control for each of the bacterial strains. The MBC values were determined using the serial subculture of 2.0 μL in micro-titration plates containing 100 μL of broth/well and a further incubation for 24 h. The minimum concentration that caused no visible growth was expressed as the MBC, which indicates a 99.5% demolition of the original inoculum. The optical densities of all wells were measured using a microplate manager 4.0 (Bio-Rad Laboratories, Hercules, CA, USA) at a wavelength of 655.0 nm and related to a broth medium and diluted extracts with no bacteria and to a positive control. Two positive controls were used, ampicillin (Panfarma, Belgrade, Serbia) and streptomycin (Sigma-Aldrich, Saint-Quentin-Fallavier, France), at concentrations of 69–1100 and 687–2290 µmol/mL in sterile saline, respectively. In addition, five percent aqueous solution of DMSO was used in the current test as a negative control.

### 2.7. Antifungal Test

The antifungal activity was evaluated using the following eight fungal strains: Penicillium ochrochloron, *P. verrucosum var. cyclopium*, *P. funiculosum*, *Aspergillus fumigatus*, *A. ochraceus*, *A. niger*, *A. versicolor*, and *Trichoderma viride*. The strains utilized were picked from the mycological laboratory’s stock (Institute for Biological Research, Belgrade, Serbia). The antibiotics used as positive controls in this study were streptomycin (Sigma-Aldrich, Saint-Quentin-Fallavier, France) and ampicillin (Panfarma, Belgrade, Serbia). The active ingredients in the antifungal controls employed are ketoconazole and bifonazole.

Ketoconazole is a member of the imidazole class of drugs, and it is used to treat fungal infections of the mouth (thrush), esophagus, reproductive system (yeast infection), and other areas. In 2013, the European Medicines Agency recommended that the marketing authorization of oral ketoconazole-containing medicines should be suspended in the face of hepatic risks [30]. Bifonazole inhibits the synthesis of ergosterol, a molecule that constitutes the fungal membrane.

## 3. Results and Discussions

### 3.1. FT-IR Analysis

This work presents a simple and convenient approach for the synthesis of HAp/pectin nanoparticles. Figure 2 shows the FT-IR spectra of pectin, as well as an overlay spectrum of HAp and the synthetized HAp/pectin composites. The pectin spectrum exhibits a band at 3335 cm^−1^ corresponding to the stretching of the hydroxyl group and a peak at 2929 cm^−1^ corresponding to the vibration of the CH. The carbonyl (C=O) stretching vibration of the esterified carboxyl groups is shown by the 1731 cm^−1^ peak [29,31]. The signal at 1614 cm^−1^ could be attributable to carboxyl carbonyl stretching vibration. The FT-IR spectra of the HAp nanoparticles is dominated by the typical PO_4_ bands: asymmetric mode at 1046–1087 cm^−1^, symmetric stretching mode at 962 cm^−1^, and bending mode at 600–650 cm^−1^; the bands at 3572 and 632 cm^−1^ belong to the vibration of hydroxyl (O–H) group, the bands at 1089, 1045, and 962 cm^−1^ are the characterization of phosphate stretching vibration, and the bands observed at 601 and 570 cm^−1^ are due to the phosphate being in vibration. 

The vibration band at 1400–1450 cm^−1^ could be attributed to the C–H plane deformation vibrations of alkyl and aryl methyl, methylene, and methoxy groups. The in-plane bending vibration of C–O–H is responsible for the 1419 cm^−1^ peak. The existence of methoxy groups is indicated by the presence of the C–O–C asymmetric stretching vibration at 1251 cm^−1^ (OCH_3_). Multiple peaks in the 1000–1200 cm^−1^ range could be attributed to the vibration of the C–C and C–O bonds of the glycosidic linkage and pyranoid ring. Multiple bands at 1049, 1082, and 1151 cm^−1^ in the pectin FT-IR spectra are features of pectin polysaccharides and can be related to the vibration of –C–C–, –C–OH, and –C–O–, respectively (24). Furthermore, O–C–O bending and CH deformation were observed in the 400–800 cm^−1^ range. Figure 2 shows the FT-IR spectra of HAp and HAp/pectin nanocomposite. The apatite network group PO_4_^3−^ absorptions reveal bands at 1090–1047, 962, 602, and 572 cm^−1^. The comparison of the three FT-IR spectra clearly demonstrates a shift in the 1047 cm^-1^ vibration of HAp’s PO_4_^3−^. The stretching of the hydroxyl group and the C–O bands, as well as the presence of the PO4^3−^ band at 1026 cm^−1^, confirm the interaction between the two components and the development of the composite. The new vibration bands corresponding to the C–O in the composites are 1038 and 1035 cm^−1^ for the 50/50 and 90/10 HAp/pectin composites, respectively. The FT-IR results show that pectin and hydroxyapatite have a significant intermolecular interaction.

### 3.2. SEM Analysis

Figure 3 shows SEM micrographs of pectin, HAp, and HAp/pectin composites. The dispersion of HAp into the pectin matrix can be easily spotted. As shown in Figure 3A,B, the HAp and pectin exhibit a spherical and semispherical morphology. Composites, on the other hand, have a non-regular morphology and agglomerate (Figure 3C,D).

Furthermore, when the HAp concentration is reduced and the pectin concertation is increased to 50%, the size of the composite particles reduces and the morphology appears to become more regular in shape, as illustrated in Figure 3. The SEM results demonstrate that the presence of pectin is significant since it controls the purity and shape of the composite [22].

The influence of pectin content on particle size was investigated. The results are displayed in Figure 4, which demonstrates that the size of the HAp nanoparticles grew as the concentration of pectin dropped. In the examined range, the dependence was linear. This is most likely due to the high concentration of HAp particles dispersed inside the polymer, which prevents HAp particle aggregation.

### 3.3. XRD Analysis of HAp/Pectin Composite

Figure 5 shows the X-ray patterns of HAp and nanoparticles of HAp/pectin. The HAp particles are clearly visible in Figure 5B to be in the micro size range. The XRD pattern of HAp shows that the distinctive peaks for HAp match the JCPDS card No. 09-0432.

The diffraction peaks are intense and narrow, especially in planes 002, 211, 112, and 300, showing that the HAp is crystalline. Figure 5C,D show the X-ray diffractions of HAp/pectin nanocomposites with pectin mole percentages of 10 and 50, respectively. The peak at 2 thetas = 32° in the plane 211 is a characteristic of hydroxyapatite. At 10% and 50% of pectin, this 211-peak shifted to 2 thetas = 35°. The results are an indication that a low concentration of pectin slightly impacts the crystallinity of HAp material (Figure 5), whereas at a higher concentration of pectin, the composite exhibits higher porosity.

Normally, for biomedical applications, HAp with low crystallinity is more attractive due to its high in vivo restorability [31,32]. Therefore, the presence of pectin at 50 wt.% has significantly affected the purity and reduced the crystallinity of the HAp nanoparticles produced. The amount of pectin in the composite has a direct impact on the texture and porosity of the manufactured material. There was a substantial interaction between the calcium ions of HAp and the carbonyl groups of pectin. This activity significantly aided the production of amorphous HAp/pectin nanoparticles.

Scherrer’s equations were used to calculate the crystallite size and crystallinity of the produced HAp/pectin nanocomposite (Equations (1) and (2)). Table 2 summarizes the results. Because it is sharp and isolated, the diffraction peak at 32–35°, which corresponds to plane 211, was chosen for the calculation of crystallite size and crystallinity. At low concentrations of pectin, the crystallinity of HAp was slightly affected, whereas at higher concentrations of pectin, the composite had a low crystallinity. The XRD analysis revealed that the HAp nanoparticles exhibited 10.9306% crystallinity in the absence of pectin. It was lowered to roughly 7.2986% when the concentration climbed to 10 wt.% pectin and to 7.2985% in the presence of 50 wt.% pectin.

Based on these results, Figure 6 depicts a schematic representation of the strong interaction between the calcium ions of HAp and the CO groups of pectin. The model can also be used to explain the FT-IR and SEM results. Such behavior greatly enhanced the formation of HAp/pectin nanoparticles. 

### 3.4. Total Organic Carbon Production

The total organic carbon data suggested that pectin produces more carbon than composites. The TOC data also showed that the presence of HAp reduced CO_2_ emissions, as seen in Figure 7. The TOC level of composites has been linked to changes in the HAp percentage.

### 3.5. Antibacterial Efficacy

In this part, the antibacterial activity of three products: pectin extracted from *O. ficus-indica* cladoes, HAp, and HAp/pectin composites, was evaluated (Figure 8a). Table 3 shows the acquired results.

The results reported in Table 3 reveal that the HAp/pectin composites (50% and 50% pectin) had much higher efficacy against Gram-positive and Gram-negative bacteria than neat pectin and hydroxyapatite. The antimicrobial activity is at least twice that obtained by the streptomycin antibiotic, and about five times as strong as ampicillin.

Microbial biofilms have raised severe issues in the healthcare, medical, and food industries due to their inherent resistance to standard drugs and cleaning procedures, as well as their capacity to firmly adhere to surfaces for persistent contamination [33,34]. In this regard, additional experiments are highly urged to examine the antibacterial activity of composites against bacterial growth inhibition and to demonstrate antibacterial capabilities against biofilms.

### 3.6. Antifungal Effect 

The antifungal efficacy of three materials was tested: pectin extracted from *O. ficus-indica* rackets, HAp, and the HAp/pectin nanocomposite (90/10). Table 4 reveals that the antifungal efficacy of the HAp/pectin composite greatly exceeds that of HAp. It is also relatively more effective than ketoconazole against the two fungi (*P. funiculosum* and *T. viride*) and even showed better activity than bifonazole against the three fungi (*A. fumigatus*, *A. versicolor*, and *A. ochraceus*).

Polymers with intrinsic antimicrobial activities have long been an intriguing research area, owing to their widespread natural abundance in materials such as chitin, pectin, chitosan, and carrageen, as well as the fact that they can be adhered to surfaces without losing their antimicrobial activities [35]. In a recent study, the antimicrobial activity of chitosan-based films incorporated of betel leaf extract (BE) was investigated [36]. The authors demonstrated that films containing BE with different concentrations (5, 10, and 15%) had an enhanced inhibitory activity against the growth of foodborne microorganisms. Other researchers demonstrated that the combination of nanofibrillar cellulose with Ag nanoparticles presented improved antibacterial activity against *S. aureus* and antifungal activity against *Candida albican* [37,38]. Another study revealed that styrene-maleic anhydride copolymer showed also an excellent bactericidal activity against *E. coli* and *S. aureus* even though their antifungal activity against *A. niger* was not satisfactory [39]. Our findings demonstrate the potential of HAp/pectin nanocomposite as antibacterial and antifungal material, highlighting its future application as a novel drug-free antimicrobial copolymer.

## 4. Conclusions

In this study, the HAp nanoparticles were successfully converted to a biomaterial nanocomposite with pectin by the double decomposition method. The particle size of HAp/pectin nanoparticles increased when the pectin-to-HAp ratio decreased. Furthermore, the presence of pectin at low concentrations promotes the formation of amorphous regions in the composite, and showed a noticeable influence on the prepared nanomaterial texture and porosity, helping in producing amorphous HAp/pectin nanoparticles. In the absence of pectin, the XRD analysis revealed that the HAp nanoparticles had 10.93% crystallinity. When the pectin concentration reached 10 wt.%, it was reduced to approximately 7.29%. In addition to strong antifungal activity, the synthesized composite displayed powerful antibacterial efficacy against Gram-positive (*S. aureus*, *B. cereus*) and Gram-negative (*E. coli* and *P. aeruginosa*) bacteria. Its considerable antimicrobial impact suggests that it could be used in the food and pharmaceutical industries.

## Figures and Tables

**Figure 1 polymers-14-04446-f001:**
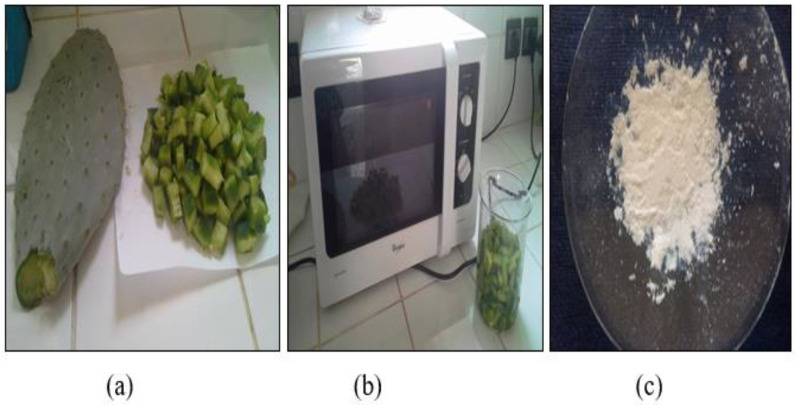
Pectin extraction from prickly pear cladodes (**a**); Microwave device (**b**); Pectin powder (**c**).

**Figure 2 polymers-14-04446-f002:**
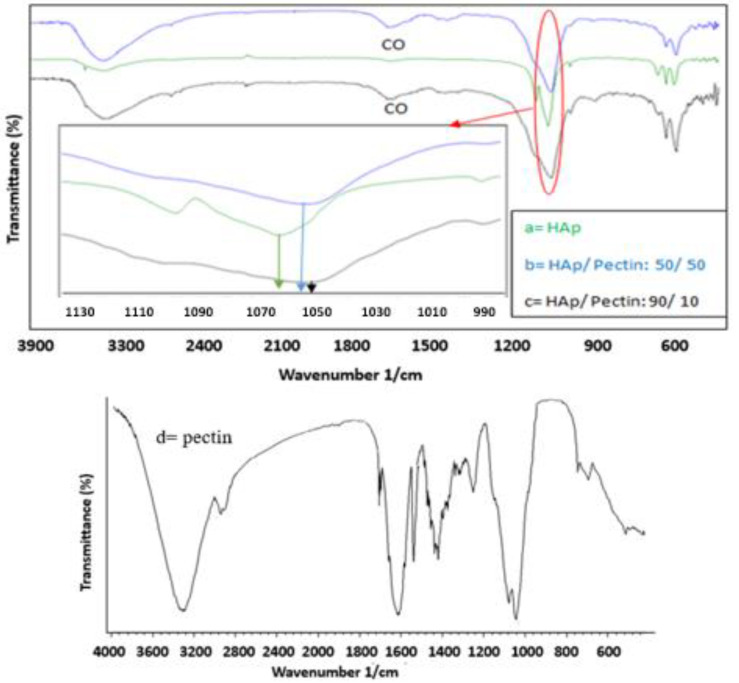
FT-IR spectra of the synthesized HAp nanoparticles in the presence and absence of various pectin concentrations, with molar ratios of 100/0 (a), 50/50 (b), 90/10 (c), and pectin (d) extracted from the prickly pear cladodes.

**Figure 3 polymers-14-04446-f003:**
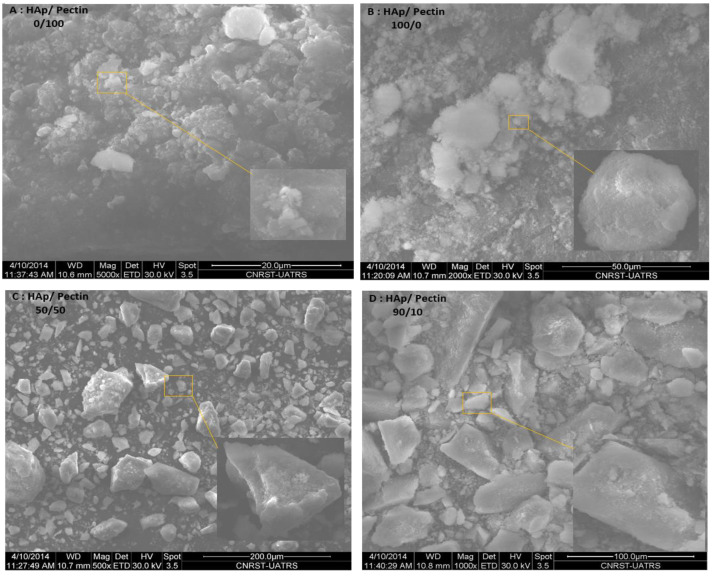
(**A**) SEM micrographs of pectin, (**B**) HAp, (**C**) HAp/pectin/50/50, and (**D**) HAp/pectin90/10 nanoparticles.

**Figure 4 polymers-14-04446-f004:**
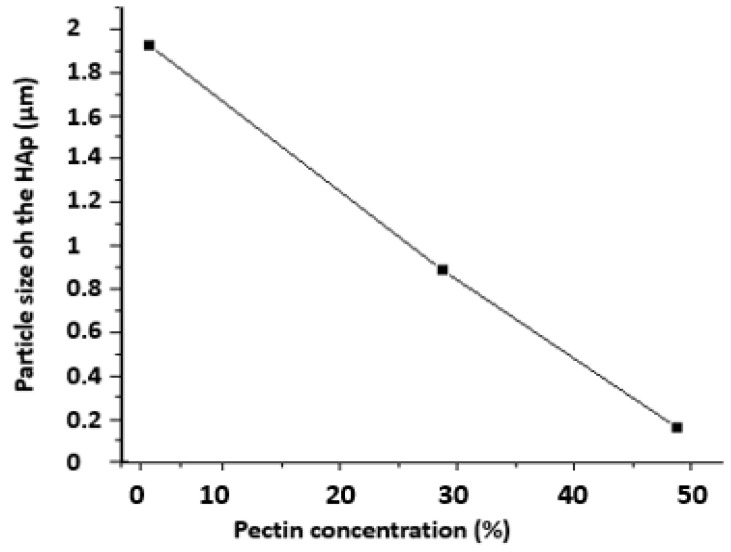
Particle Size of HAp/Pectin Nanoparticles.

**Figure 5 polymers-14-04446-f005:**
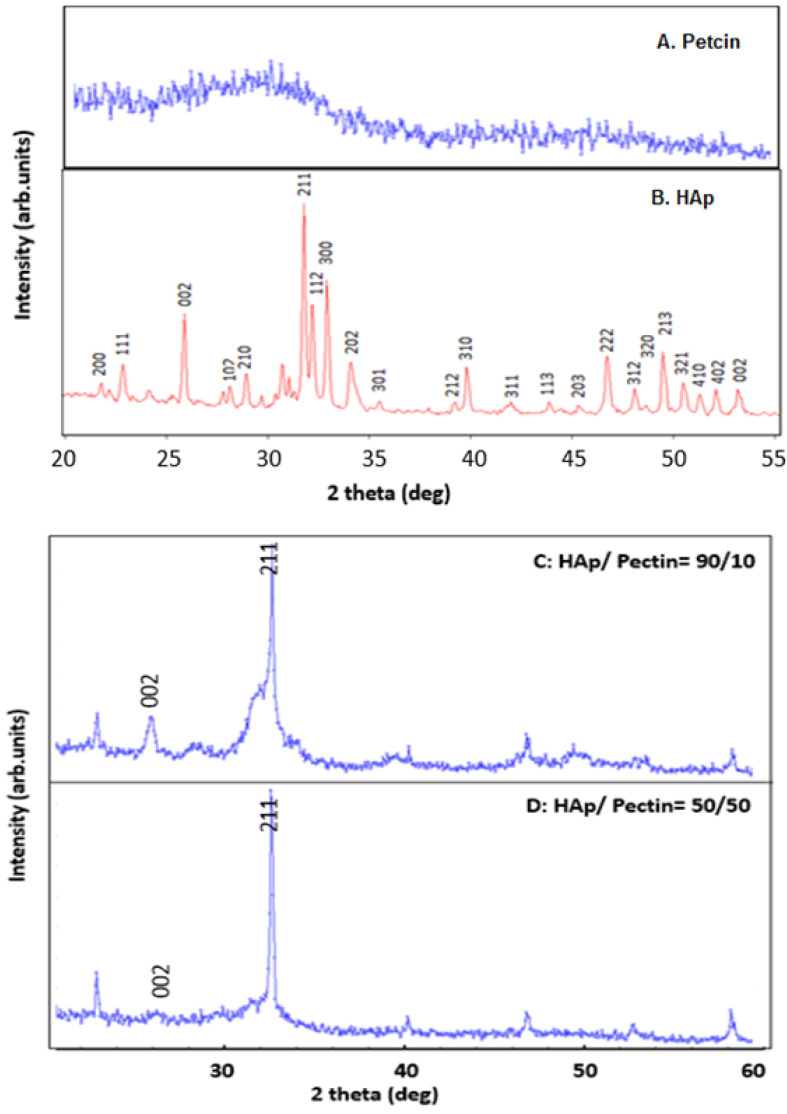
(**A**) XRD of pectin, (**B**) HAp, (**C**) HAp/pectin90/10, and (**D**) HAp/pectin50/50 nanoparticles.

**Figure 6 polymers-14-04446-f006:**
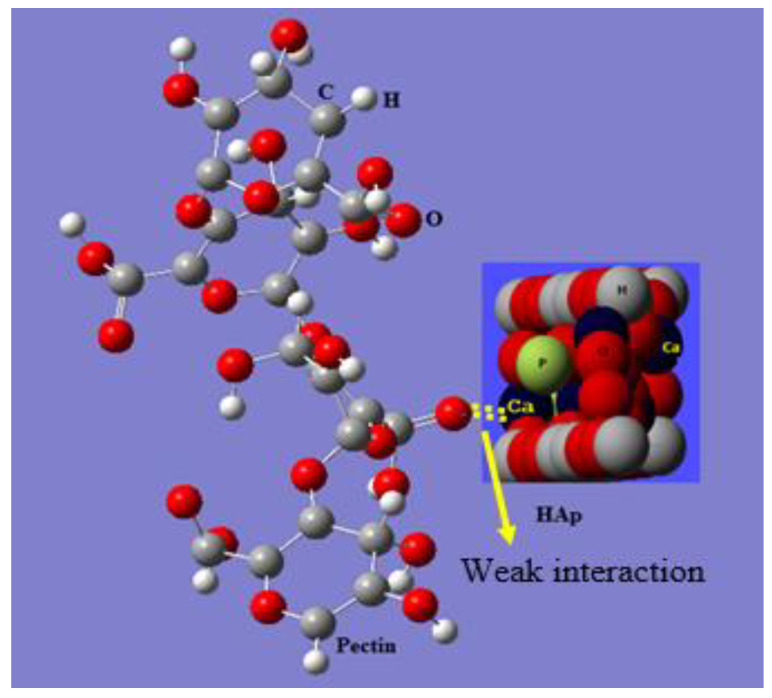
Schematic Model of the weak interaction between the CO-pectin and Ca-HAp.

**Figure 7 polymers-14-04446-f007:**
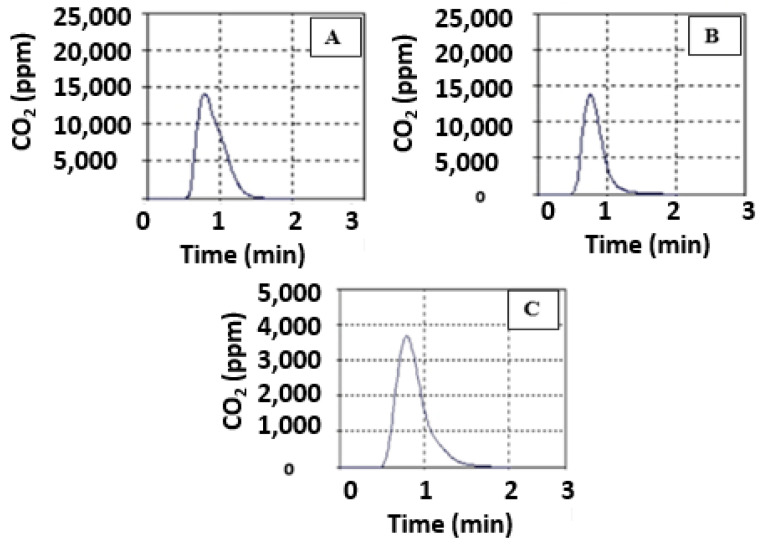
The plot of TOC vs. time for pectin and HAp/pectin composite; (**A**) = pectin, (**B**) = 90/10, and (**C**) = 50/50.

**Figure 8 polymers-14-04446-f008:**
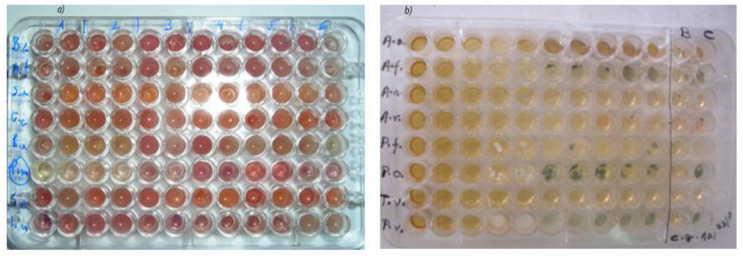
Antibacterial (**a**) and antifungal (**b**) effects of samples on different bacteria and fungi strains.

**Table 1 polymers-14-04446-t001:** List of bacterial strains studied.

Species		Gram
*Entero bactercloacae* (Clinical isolate)	-	Negative
*Salmonella typhimurium*	ATCC 13311	
*Escherichia coli*	ATCC 35210	
*Pseudomonas aeruginosa*	ATCC 27853	
*Micrococcus flavus*	ATCC 10240	Positive
*Listeria monocytogenes*	NCTC 797	
*Staphylococcus aureus*	ATCC 6538	
*Bacillus cereus* (Clinical isolate)	-	

**Table 2 polymers-14-04446-t002:** Crystallinity and crystallite size of the synthesized HAp/pectin composite.

Pectin Concentration	Plane	θ	FWHM (°)	Xc (nm)	Xs (%)
**0**	211	16.1089	0.1319	6.0241	10.9306
**10**	211	16.3294	0.1978	1.7863	7.2986
**50**	211	16.31214	0.1978	1.7863	7.2985

**Table 3 polymers-14-04446-t003:** Antibacterial activity of *Opuntia fi**cus-indica* pectin, hydroxyapatite, and composites (HAp/pectin) in mg/mL.

		*S. Aureus*	*B. Cereus*	*L. Monocytogenes*	*M. Flavus*	*P. Aeruginosa*	*E. Coli*	*S. Typhimurium*	*En. Cloacae*
**Pectin**	MIC	0.56	0.56	0.28	0.56	0.56	0.56	0.28	0.56
	MBC	1.14	1.14	1.14	1.14	1.14	1.14	1.14	1.14
**Hydroxyapatite** **(HAp)**	MIC	0.56	0.28	0.56	0.56	0.28	0.56	0.56	0.28
	MBC	1.14	0.56	1.14	1.14	0.56	1.14	1.14	0.56
**HAp/pectin** **(90%/10%)**	MIC	0.14	0.14	0.56	0.28	0.14	0.28	0.56	0.28
	MBC	0.28	0.28	1.14	0.56	0.28	0.56	1.14	0.56
**HAp/pectin** **(50%/50%)**	MIC	0.14	0.14	0.28	0.28	0.14	0.28	0.28	0.28
	MBC	0.28	0.28	0.56	0.56	0.28	0.56	0.56	0.56
**Streptomycin**	MIC	0.04	0.09	0.17	0.17	0.34	0.26	0.17	0.17
	MBC	0.09	0.17	0.34	0.34	0.68	0.52	0.34	0.34
**Ampicillin**	MIC	0.25	0.25	0.25	0.37	0.74	0.37	0.37	0.25
	MBC	0.37	0.37	0.37	0.49	1.24	0.74	0.49	0.49

**Table 4 polymers-14-04446-t004:** Antifungal activity of *Opuntia ficus-indica* pectin, hydroxyapatite, and composites (HAp/pectin) in mg/mL.

		*A. Fumigatus*	*A. Versicolor*	*A. Ochraceus*	*A. Niger*	*P. Ochrochloron*	*P. Funiculosum*	*P. Verrucosum*	*T. Viride*
**Pectin**	MIC	0.55	1.10	0.55	1.10	NA	0.14	0.28	0.55
	MFC	1.10	1.10	1.10	1.10	NA	0.28	0.55	1.10
**Hydroxyapatite** **(HAp)**	MIC	0.19	0.19	0.39	0.78	0.19	0.19	0.39	0.39
	MFC	0.38	0.38	0.78	1.55	0.39	0.39	0.78	0.78
**HAp/pectin** **(90/10)**	MIC	0.098	0.098	0.048	0.39	NA	0.19	NA	0.39
	MFC	0.19	0.19	0.098	0.78	NA	0.39	NA	0.78
**Ketoconazole**	MIC	0.20	0.20	0.15	0.20	1.00	0.20	0.20	1.00
	MFC	0.50	0.50	0.20	0.50	1.50	0.50	0.30	1.50
**Bifonazole**	MIC	0.15	0.10	0.15	0.15	0.20	0.20	0.10	0.15
	MFC	0.20	0.20	0.20	0.20	0.25	0.25	0.20	0.20

## Data Availability

Not applicable.

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
