# Peer review of "Design of Nanohydroxyapatite/Pectin Composite from Opuntia Ficus-Indica Cladodes for the Management of Microbial Infections"

_polymers, 2022, doi:10.3390/polym14204446_

Round 1

Reviewer 1 Report

The article entitled “Design of Nanohydroxyapatite/Pectin Composite from Opuntia ficus-indica Cladodes for the Management of Microbial Infections” was carefully reviewed.

The authors did extensive research on the topic and presented work that is interesting.

 Still it needs revision before considering publication in Polymers.

 Limitation of the work, novelty and contributions should be highlighted more.

 The introduction still needs to be improved. Following recent relevant references should be included in the introduction part for more readable; about various nanomaterials for biomedical applications.

Polymers 14(19), 4128, 2022. Journal of microbiological methods 159, 200-210, 2019. Polymers for Advanced Technologies 33 (9), 2872-2885, 2022.  

 The purification of the prepared nanomaterials can be explained properly.

 The provided FTIR peak intensity is very poor, the author should provide NMR data for the confirmation.

To confirm the functional groups authors should read the below articles and discuss properly in the result and discussion section.

Reactive and Functional Polymers 67 (6), 503-514, 2007.

For the antimicrobial study the following relevant references can be referred for comparison study.

Green Materials 9 (2), 49–68, 2021.  Journal of microbiological methods 163, 105650, 2019.

 The provided SEM images need full scale with high magnification details.

 Still, there are some typographical errors. Therefore, the authors are advised to recheck the whole manuscript.

Overall the article is fairly well written, after addressing the above comments the article may be considered for publication.

Author Response

Dear Reviewer,

We would like to thank you very much for your valuable comments. We answered your comments and we added the suggested references. I am sure with your comments the paper looks much better.

Thanks

Open Review

English language and style

( ) Extensive editing of English language and style required
(x) Moderate English changes required
( ) English language and style are fine/minor spell check required
( ) I don't feel qualified to judge about the English language and style

Yes

Can be improved

Must be improved

Not applicable

Does the introduction provide sufficient background and include all relevant references?

( )

(x)

( )

( )

Are all the cited references relevant to the research?

( )

(x)

( )

( )

Is the research design appropriate?

( )

(x)

( )

( )

Are the methods adequately described?

( )

(x)

( )

( )

Are the results clearly presented?

( )

(x)

( )

( )

Are the conclusions supported by the results?

( )

(x)

( )

( )

Comments and Suggestions for Authors

The article entitled “Design of Nanohydroxyapatite/Pectin Composite from Opuntia ficus-indica Cladodes for the Management of Microbial Infections” was carefully reviewed.

The authors did extensive research on the topic and presented work that is interesting.

 Still it needs revision before considering publication in Polymers.

 Limitation of the work, novelty and contributions should be highlighted more.

-  The introduction still needs to be improved. Following recent relevant references should be included in the introduction part for more readable; about various nanomaterials for biomedical applications.

Polymers have been extensively used as nanocarriers in recent decades [24,34,36].  The antibacterial polymers usually obtained either by synthesis of monomeric biocide moiety and subsequent polymerization or copolymerization with another monomer or modification can also be brought by grafting of N-alkylated poly (4-vinylpyridine) quarternized polyethyleneimine and quaternary derivatives of acrylic acid onto numerous materials such as cellulose [23]. Others studies demonstrate that ZnO/Ag nanocomposites, membranes loaded with HAp-Ag3PO4 nanoparticles, and silver nanoparticle–activated carbon composite nanofiber membranes exhibit an enhanced antimicrobial activity [17,18,33].

Both HAp and pectin have the functionality required for strong interaction and compatibility represented by H-bonding. The main objective of the present study is to prepare a novel biocomposite of HAp and pectin, obtained from O. ficus-indica. A process for making nanoparticles is offered in this study. FT-IR, XRD, and SEM studies were used to characterize the synthetized nanocomposite. The HAP/pectin nanocomposite performed a dual function in water purification. It exhibited a high affinity for a variety of antimicrobial activity against both Gram-negative and Gram-positive bacteria, and fungi, providing a new outlook for the development of antimicrobial agents.

-Polymers 14(19), 4128, 2022. Journal of microbiological methods 159, 200-210, 2019. Polymers for Advanced Technologies 33 (9), 2872-2885, 2022.  

I added the references in the text

 - The purification of the prepared nanomaterials can be explained properly.

At room temperature, an aqueous solution of pectin polymer was made and labeled as solution A. A prepared HAp sample was disseminated in dimethylformamide and labeled solution B. The two solutions, were then combined. After approximately 1.5 h, an opaque milky white suspension was generated. Over a 30-min, the temperature of the generated suspension was gradually elevated at a rate of 2°C/min to 50 °C. To get a homogeneous suspension, the temperature was kept at 50 °C for 2 h. The nanoparticles in the suspension were centrifuged and washed twice with ethanol to eliminate unreacted material, potential contaminants, and to speed drying. HAp/pectin nanocomposites were synthetized with molar ratios of 100/0, 50/50 and 90/10.

 - The provided FTIR peak intensity is very poor, the author should provide NMR data for the confirmation.

The FT-IR spectra of the HAp nanoparticles is dominated by the typical PO4 bands: asymmetric mode at 1046–1087 cm−1, symmetric stretching mode at 962 cm−1 and bending mode at 600–650 cm−1, the bands at 3572 and 632 cm−1 belong to the vibration of hydroxyl (O–H) group, the bands at 1089, 1045 and 962 cm−1 are the characterization of phosphate stretching vibration and the bands observed at 601, 570 cm−1 are due to the phosphate being in vibration.

For the NMR unfortunately, we do not have one. Also, there is no money to cover this type of analysis but we used this reference when we confirmed our analysis.

Christian Jager,  Thea Welzel, Wolfgang Meyer-Zaika and Matthias Epple A solid-state NMR investigation of the structure of nanocrystalline hydroxyapatite  Magn. Reson. Chem. 2006; 44: 573–580

- To confirm the functional groups authors should read the below articles and discuss properly in the result and discussion section.

Reactive and Functional Polymers 67 (6), 503-514, 2007.

For the antimicrobial study the following relevant references can be referred for comparison study.

Green Materials 9 (2), 49–68, 2021.  

Journal of microbiological methods 163, 105650, 2019.

I added the references in the text

 The provided SEM images need full scale with high magnification details.

Figure 3. SEM micrographs of pectin, HAp, and pectin/HAp nanoparticles.

-  Still, there are some typographical errors. Therefore, the authors are advised to recheck the whole manuscript.

I made the corrections

Overall the article is fairly well written, after addressing the above comments the article may be considered for publication.

Reviewer 2 Report

The manuscript “Design of Nanohydroxyapatite/Pectin Composite from Opuntia ficus-indica Cladodes for the Management of Microbial Infections” deals with the production of novel nanocomposite biomaterials based on hydroxyapatite and pectin, using the double decomposition method. In general, the work is interesting; however, the quality of presentation and discussion of the results has to be improved.

Detailed comments:

- Use journal template and reference style.

- Abstract. Add quantitative results to this section.

- Introduction. Enlarge the state of the art adding other relevant and recent works in the field, as for instance: Baldino et al., Journal of Chemical Technology and Biotechnology, 2019, 94(1), pp. 98–108; Liu et al., Frontiers in Materials, 2022, 9, 894451; Cuadra et al., Applied Sciences, 2022, 12(10), 5023; etc.

- The novelty of the work is not clear from the Introduction. Rewrite the last paragraph.

- M&M. Improve the description of the various preparation procedures; e.g., “A and B, the

two solutions, were combined.”, specify in which ratio were mixed, etc.

- R&D. Improve the quality and resolution of figures. In particular, improve the quality of SEM images. The results have to be better discussed and the simple addition of refs is not enough.

- Conclusions are a summary of the work. Rewrite in a more critical way.

- Reduce self-citations and enlarge the study of the literature.

Author Response

Dear Reviewer,

We would like to thank you very much for your valuable comments. We answered your comments and we added the suggested references. I am sure with your comments the paper looks much better.

Thanks

Open Review

English language and style

( ) Extensive editing of English language and style required
(x) Moderate English changes required
( ) English language and style are fine/minor spell check required
( ) I don't feel qualified to judge about the English language and style

Yes

Can be improved

Must be improved

Not applicable

Does the introduction provide sufficient background and include all relevant references?

( )

( )

(x)

( )

Are all the cited references relevant to the research?

( )

( )

(x)

( )

Is the research design appropriate?

( )

(x)

( )

( )

Are the methods adequately described?

( )

( )

(x)

( )

Are the results clearly presented?

( )

( )

(x)

( )

Are the conclusions supported by the results?

( )

( )

(x)

( )

Comments and Suggestions for Authors

The manuscript “Design of Nanohydroxyapatite/Pectin Composite from Opuntia ficus-indica Cladodes for the Management of Microbial Infections” deals with the production of novel nanocomposite biomaterials based on hydroxyapatite and pectin, using the double decomposition method. In general, the work is interesting; however, the quality of presentation and discussion of the results has to be improved.

Detailed comments:

- Use journal template and reference style.

i did the new paper style

- Abstract. Add quantitative results to this section.

We further assess the antibacterial and antifungal activity of the developed nanocomposite against a number of pathogenic bacteria and fungi, evaluated by the well diffusion method. In the absence of pectin, the XRD analysis revealed that the HAp nanoparticles had 10.93% crystallinity. When the pectin concentration reached 10 wt.%, it was reduced to approximately 7.29%. All synthesized nanocomposites demonstrated strong antimicrobial activity against both Gram-positive (Staphylococcus aureus and Bacillus cereus) and Gram-negative (Escherichia coli and Pseudomonas aeruginosa) bacteria in addition to various fungi (e.g., Aspergillus fumigatus, Penicillium funiculosum, and Trichoderma viride). This study endorses the Hap/Pectin nanocomposite as an efficient antimicrobial material for biomedical advanced applications.

- Introduction. Enlarge the state of the art adding other relevant and recent works in the field, as for instance: Baldino et al., Journal of Chemical Technology and Biotechnology, 2019, 94(1), pp. 98–108;

Liu et al., Frontiers in Materials, 2022, 9, 894451; Cuadra et al., Applied Sciences, 2022, 12(10), 5023; etc.

I added the references to our article

- The novelty of the work is not clear from the Introduction. Rewrite the last paragraph.

Polymers have been extensively used as nanocarriers in recent decades [24,34,36].  The antibacterial polymers usually obtained either by synthesis of monomeric biocide moiety and subsequent polymerization or copolymerization with another monomer or modification can also be brought by grafting of N-alkylated poly (4-vinylpyridine) quarternized polyethyleneimine and quaternary derivatives of acrylic acid onto numerous materials such as cellulose [23]. Others studies demonstrate that ZnO/Ag nanocomposites, membranes loaded with HAp-Ag3PO4 nanoparticles, and silver nanoparticle–activated carbon composite nanofiber membranes exhibit an enhanced antimicrobial activity [17,18,33].

Both HAp and pectin have the functionality required for strong interaction and compatibility represented by H-bonding. The main objective of the present study is to prepare a novel biocomposite of HAp and pectin, obtained from O. ficus-indica. A process for making nanoparticles is offered in this study. FT-IR, XRD, and SEM studies were used to characterize the synthetized nanocomposite. The HAP/pectin nanocomposite performed a dual function in water purification. It exhibited a high affinity for a variety of antimicrobial activity against both Gram-negative and Gram-positive bacteria, and fungi, providing a new outlook for the development of antimicrobial agents.

- M&M. Improve the description of the various preparation procedures; e.g., “A and B, the

two solutions, were combined.”, specify in which ratio were mixed, etc.

At room temperature, an aqueous solution of pectin polymer was made and labeled as solution A. A prepared HAp sample was disseminated in dimethylformamide and labeled solution B. The two solutions, were then combined. After approximately 1.5 h, an opaque milky white suspension was generated. Over a 30-min, the temperature of the generated suspension was gradually elevated at a rate of 2°C/min to 50 °C. To get a homogeneous suspension, the temperature was kept at 50 °C for 2 h. The nanoparticles in the suspension were centrifuged and washed twice with ethanol to eliminate unreacted material, potential contaminants, and to speed drying. HAp/pectin nanocomposites were synthetized with molar ratios of 100/0, 50/50 and 90/10

- R&D. Improve the quality and resolution of figures. In particular, improve the quality of SEM images. The results have to be better discussed and the simple addition of refs is not enough.

Figure 3. SEM micrographs of pectin, HAp, and pectin/HAp nanoparticles.

- Conclusions are a summary of the work. Rewrite in a more critical way.

In this study, the HAp nanoparticles were successfully converted to a biomaterial nanocomposite with pectin by the double decomposition method. The particle size of HAp/pectin nanoparticles increased when the pectin-to-HAp ratio decreased. Furthermore, the presence of pectin at low concentrations promotes the formation of amorphous regions in the composite, and showed a noticeable influence on the prepared nanomaterial texture and porosity, helping in producing amorphous HAp/pectin nanoparticles. . In the absence of pectin, the XRD analysis revealed that the HAp nanoparticles had 10.93% crystallinity. When the pectin concentration reached 10 wt.%, it was reduced to approximately 7.29%. In addition to strong antifungal activity, the synthesized composite displayed powerful antibacterial efficacy against gram-positive (S. aureus, B. cereus) and gram-negative (E. coli and P. aeruginosa) bacteria. Its considerable antimicrobial impact suggests that it could be used in the food and pharmaceutical industries.

- Reduce self-citations and enlarge the study of the literature.

I did

Submission Date

18 September 2022

Date of this review

03 Oct 2022 23:37:27

Round 2

Reviewer 1 Report

The revised entitled "Design of Nanohydroxyapatite/ Pectin Composite from Opuntia ficus-indica Cladodes for the Management of Microbial Infections" was carefully reviewed. 

The authors have made the suggested corrections carefully.

Hence, I recommend this article may be acceptable for publication in the present form.

Reviewer 2 Report

The authors performed the modifications proposed by the Reviewer. However, references style has to be checked, following the journal template.